# HS1BP3 negatively regulates autophagy by modulation of phosphatidic acid levels

Petter Holland[1,*], Helene Knævelsrud[1,*], Kristiane Søreng[1], Benan J. Mathai[1], Alf Håkon Lystad[1], Serhiy Pankiv[1], Gunnveig T. Bjørndal[1], Sebastian W. Schultz[2], Viola H. Lobert[2], Robin B. Chan[3], Bowen Zhou[3], Knut Liestøl[4], Sven R. Carlsson[5], Thomas J. Melia[6], Gilbert Di Paolo[3] & Anne Simonsen[1]

A fundamental question is how autophagosome formation is regulated. Here we show that the PX domain protein HS1BP3 is a negative regulator of autophagosome formation. HS1BP3 depletion increased the formation of LC3-positive autophagosomes and degradation of cargo both in human cell culture and in zebrafish. HS1BP3 is localized to ATG16L1- and ATG9-positive autophagosome precursors and we show that HS1BP3 binds phosphatidic acid (PA) through its PX domain. Furthermore, we find the total PA content of cells to be significantly upregulated in the absence of HS1BP3, as a result of increased activity of the PA-producing enzyme phospholipase D (PLD) and increased localization of PLD1 to ATG16L1-positive membranes. We propose that HS1BP3 regulates autophagy by modulating the PA content of the ATG16L1-positive autophagosome precursor membranes through PLD1 activity and localization. Our findings provide key insights into how autophagosome formation is regulated by a novel negative-feedback mechanism on membrane lipids.

[1] Department of Molecular Medicine, Institute of Basic Medical Sciences, University of Oslo, PO Box 1112, 0317 Oslo, Norway. [2] Centre for Cancer Biomedicine, Faculty of Medicine and Department of Molecular Cell Biology, Institute for Cancer Research, Oslo University Hospital, 0379 Oslo, Norway. [3] Department of Pathology and Cell Biology, Taub Institute for Research on Alzheimer's Disease and the Aging Brain, Columbia University Medical Center, 630 West 168th Street, New York, New York 10032, USA. [4] Centre for Cancer Biomedicine, Faculty of Medicine, University of Oslo, 0379 Oslo, Norway. [5] Department of Medical Biochemistry and Biophysics, Umeå University, SE-901 87 Umeå, Sweden. [6] Department of Cell Biology, Yale University School of Medicine, PO Box 208002, 333 Cedar Street, New Haven, Connecticut 06520–800210032, USA. * These authors contributed equally to this work. Correspondence and requests for materials should be addressed to A.S. (email: anne.simonsen@medisin.uio.no).

Autophagy targets intracellular components for lysosomal degradation to promote cellular and organismal health and homoeostasis, and has been shown to protect against neurodegeneration and cancer, help remove invading pathogens and promote longevity[1]. Macroautophagy (here referred to as autophagy) is characterized by the formation of double-membrane autophagosomes from an expanding cargo-enwrapping phagophore and the subsequent fusion of autophagosomes with lysosomes. Autophagy is induced by stresses like starvation and also provides cellular quality control under basal conditions[2]. Autophagy must be tightly controlled at each step of the process; autophagosome formation without proper turnover is linked to neurodegenerative disorders such as Alzheimer's disease[3], defective as well as excessive autophagy is detrimental for muscle health[4] and uncontrolled autophagy could potentially harm or even kill an otherwise healthy cell.

Nucleation of a phagophore and biogenesis of a functional autophagosome is regulated by several multi-subunit complexes, including the ULK1 complex, the integral membrane protein mATG9 and its associated proteins, the class III phosphatidylinositol (PI) 3-kinase (PI3K) complex and two ubiquitin-like conjugation systems, resulting in the conjugation of ATG12 to ATG5 and ATG8/LC3 family members to phosphatidylethanolamine (PE)[5]. ATG5–ATG12 further associates with ATG16L1 and the resulting complex is recruited to endoplasmatic reticulum-associated PI(3)P-rich sites of phagophore nucleation (called omegasomes)[6] by the PI(3)P-binding protein WIPI2 (ref. 7). Further expansion of the phagophore to generate an autophagosome requires input from several membrane sources, including the endoplasmatic reticulum[8–10], mitochondria[9,11], plasma membrane[12] and recycling endosomes[13–16]. Recycling endosome-derived membranes are positive for ATG9 and ATG16L1, and essential for autophagosome formation[13–16].

The autophagic pathway involves lipids as signalling molecules, constituents and cargo of autophagosomes. However, the role of different lipids in autophagy is not clear[17,18]. PA was initially found to activate mammalian target of rapamycin (mTOR)[19], a well-known inhibitor of autophagy, in a PLD1-specific manner[20]. Recent studies have also implicated PLD1-generated PA in autophagosome formation[21,22] and in autophagosome–lysosome fusion[23]. PI(3)P, the lipid product of the class III PI3K complex, has a central role in autophagy and several PI(3)P-binding proteins in autophagy have been identified[17,24], including the FYVE domain proteins DFCP1, a marker for omegasomes[6], the scaffold protein ALFY that links cargo to the autophagic machinery for selective autophagy[25,26] and FYCO1, which is involved in trafficking of autophagosomes on microtubuli[27]. Furthermore, the WD-repeat protein WIPI2 also binds PI(3)P and is found at omegasomes[28].

Another group of phosphoinositide-binding proteins are the PX domain-containing proteins, but little is known about their involvement in autophagy. Here we show that the PX domain protein HS1BP3 negatively regulates autophagosome formation, PA levels and PLD activity. HS1BP3 binds PA through its PX domain, which leads to the recruitment of HS1BP3 to PLD1- and ATG16L1-positive autophagosome precursor membranes. We propose that HS1BP3, through its binding to PA and inhibition of PLD1 activity, provides a novel negative-feedback mechanism to ensure the proper regulation of autophagosome biogenesis.

## Results

### HS1BP3 is a negative regulator of autophagy.
To identify PX domain proteins involved in autophagy, we recently performed an imaging-based short interfering RNA (siRNA) screen in HEK GFP-LC3 cells[13] and one of the candidate proteins was HS1BP3.

Using the individual siRNA oligos from the screen, we find that depletion of HS1BP3 results in increased amounts of GFP-LC3 spots (autophagosomes) both in complete (fed) and nutrient-deplete (starved) medium in correlation with knockdown levels (Fig. 1a–c). Depletion of HS1BP3 also increases the total intensity of endogenous LC3 spots in starved cells (Supplementary Fig. 1a).

Since depletion of HS1BP3 increased the number of autophagosomes, we next investigated whether this is due to the increased formation or inhibited maturation and turnover of autophagosomes. To this end, cells were starved in the presence or absence of the lysosomal proton pump inhibitor Bafilomycin A1 (BafA1; which inhibits autophagosome maturation and lysosomal degradation) and autophagic flux monitored by quantification of the level of PE-conjugated LC3 (LC3-II)[29]. We find that LC3-II levels are significantly increased in HS1BP3-depleted cells both in complete medium and after starvation (Fig. 1d,e), and increase further in the presence of BafA1, indicating that autophagosome formation is increased on HS1BP3 depletion. As expected, LC3 lipidation is strongly inhibited in ULK1-depleted cells. We verified that LC3 messenger RNA (mRNA) levels are not significantly affected by HS1BP3 depletion (Supplementary Fig. 1b).

To further determine if depletion of HS1BP3 activates autophagy, we studied the degradation of the cargo receptor protein p62 (also known as Sequestosome-1), which is itself an autophagy substrate[30,31]. GFP-p62 expression was shut off in a stable cell line[32] and the amount of GFP-p62 remaining after starvation was measured by flow cytometry. Whereas about half of the initial GFP-p62 is degraded in control cells, GFP-p62 strongly accumulates in ULK1-depleted cells, as well as in cells treated with BafA1 (Fig. 1f). Consistent with an increase in GFP-LC3 spots and LC3 lipidation, GFP-p62 degradation increases by 20% in cells depleted of HS1BP3 (Fig. 1f), indicating increased autophagic flux. This was further confirmed by assessing the degradation of long-lived proteins, which preferentially happens through autophagy and is inhibited by the PI3K inhibitor 3-methyladenine. As shown in Fig. 1g, the release of free $^{14}$C-valine from the degradation of previously radiolabelled long-lived proteins is increased in HS1BP3-depleted cells compared with control cells both in fed and starved conditions, further indicating that HS1BP3 is a negative regulator of autophagy.

To analyse a possible role of HS1BP3 in regulation of autophagy in vivo, we employed transient silencing of Hs1bp3 in the zebrafish line Tg(CMV:EGFP-map1lc3b)[33], using a translational-blocking morpholino targeting the start site of the Hs1bp3 mRNA. The overall homology between human and zebrafish Hs1bp3 is 36%, with the PX domain being highly conserved (67% homology; Supplementary Fig. 1c). At 2 days post fertilization (dpf), abundant GFP-LC3 puncta are present in the trunk region of the morphant compared with the control embryos (Fig. 2a–d) and this difference is even more pronounced after chloroquine treatment, known to block autophagosome degradation in zebrafish[34,35]. On injection of in vitro-transcribed-capped human Hs1bp3 mRNA alongside the Hs1bp3 morpholino, we observe a partial rescue of the phenotype at 2 dpf both with and without chloroquine treatment (Fig. 2a–d; Supplementary Fig. 1d). These results suggest that autophagy is significantly elevated in Hs1bp3 morphant zebrafish at 2 dpf and that Hs1bp3 also regulates autophagy in vivo.

### HS1BP3 interacts with cortactin.
HS1BP3 was originally identified as an interaction partner of the actin cross-linking protein HS1 (ref. 36). HS1 is exclusively expressed in cells of hematopoietic lineage, whereas other cells express the homologous protein cortactin[37]. We therefore asked whether

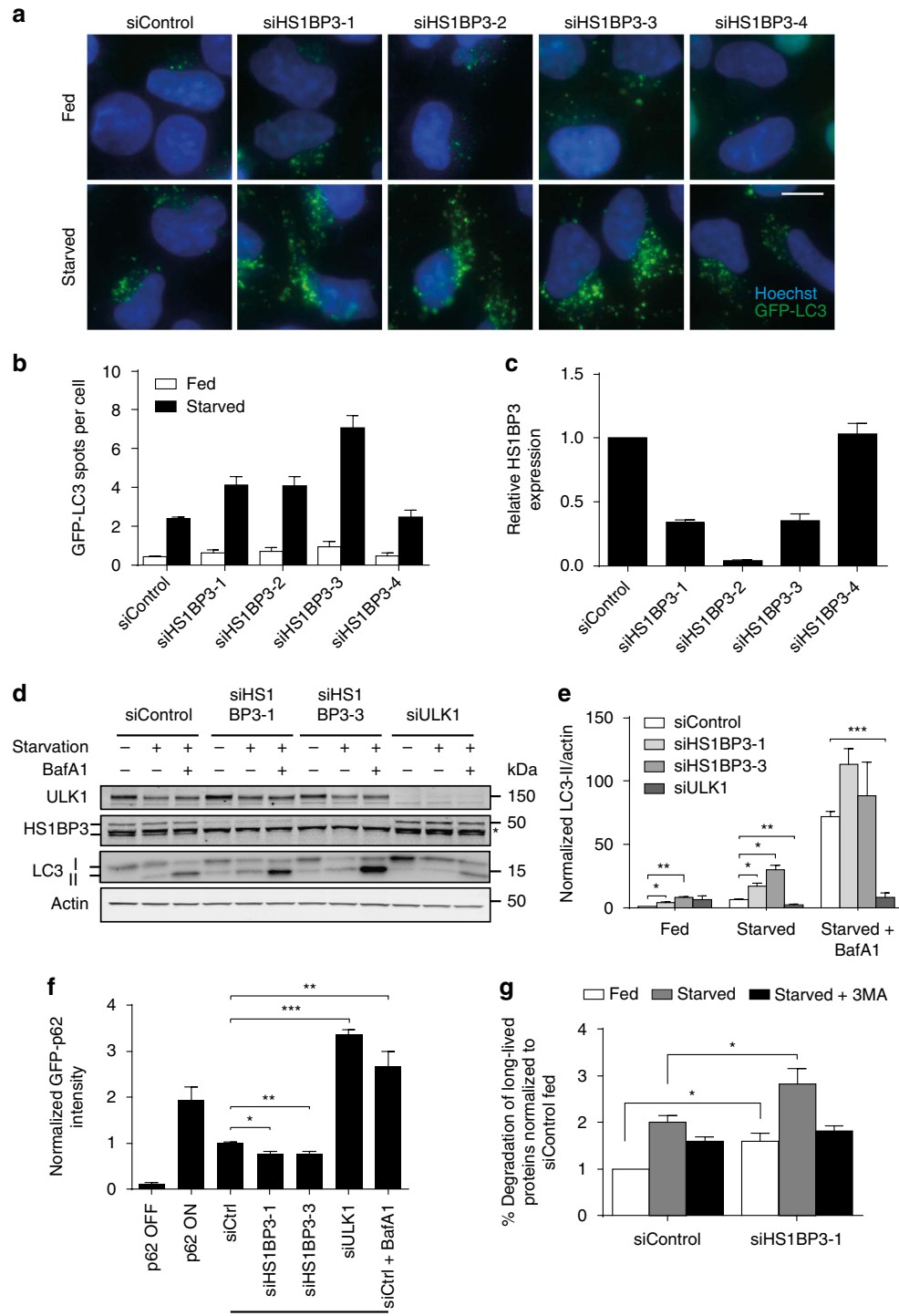

**Figure 1 | HS1BP3 is a negative regulator of autophagy.** (**a**) HEK GFP-LC3 cells were transfected with four individual siRNA oligonucleotides against HS1BP3. 72 h post transfection the cells were starved or not for 2 h in EBSS, followed by fixation and fluorescence microscopy. Scale bar, 10 μm. (**b**) The number of GFP-LC3 spots per cell in **a** was quantified by high-content analysis (mean ± s.d. from two independent experiments in triplicates, ∼50,000 cells analysed per condition). (**c**) Relative expression of HS1BP3 after siRNA knockdown was measured by quantitative PCR with reverse transcription (mean ± s.d.). (**d**) HEK GFP-LC3 cells were transfected with the indicated siRNA oligos and starved or not for 2 h in EBSS in the presence or absence of BafA1. * Indicates an unspecific band in the HS1BP3 immunoblot. (**e**) The level of LC3-II/actin was quantified from immunoblots and normalized to siControl fed (mean ± s.e.m., n = 5). (**f**) HEK GFP-p62 cells were transfected with siRNA against HS1BP3 or ULK1. Expression of GFP-p62 was induced by addition of tetracycline (compare ON versus OFF) for 48 h before expression was shut off and the cells were incubated in EBSS (starved) for 2.5 h to induce autophagic degradation of GFP-p62. GFP-p62 intensity was monitored by flow cytometry and normalized to starved siControl (siCtrl; mean ± s.e.m., n = 4). (**g**) The degradation of long-lived proteins in HeLa cells transfected with control siRNA or siRNA against HS1BP3 was quantified as the release of $^{14}$C-valine after 4 h starvation in the absence or presence of 3-methyladenine (3MA) and normalized to the degradation in fed control cells (mean ± s.e.m., n = 3). *P < 0.05, **P < 0.01, ***P < 0.001, by Student's t-test.

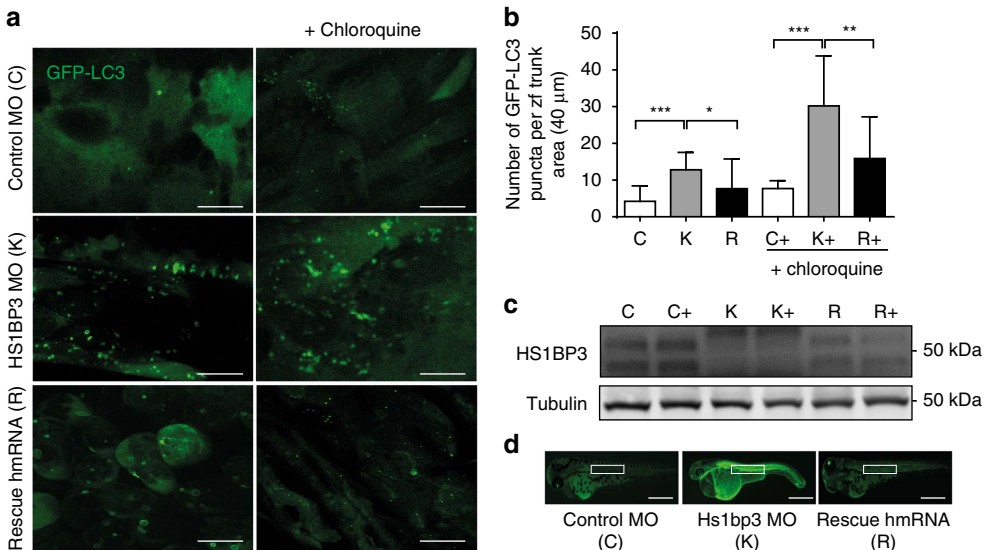

**Figure 2 | HS1BP3 regulates autophagy in zebrafish.** (**a**) Representative confocal images of GFP-LC3 puncta (autophagosomes) in the trunk area of GFP-LC3 transgenic zebrafish embryos injected with control morpholino (C), Hs1bp3 translational-blocking morpholino (K), and the human Hs1bp3 mRNA coinjected with the morpholino (R) and imaged at 2 dpf with or without pre-treatment with chloroquine (10 μM) for 6 h. Scale bars, 10 μm. (**b**) GFP-LC3 puncta were counted in the trunk region (marked in **d**) of the transgenic zebrafish embryos at 2 dpf (mean ± s.e.m., $n = 3$). Total of 7–13 embryos were used for each condition per experiment. *$P < 0.05$, **$P < 0.01$, ***$P < 0.001$, by Student's t-test. (**c**) Representative immunoblotting of Hs1bp3 and Tubulin in whole lysates of zebrafish embryos at 2 dpf, treated with or without chloroquine for 6 h before harvest. (**d**) Representative light fluorescent microscopy images of whole embryos at 2 dpf. Scale bars, 300 μm.

HS1BP3 also binds to cortactin and whether this interaction is involved in HS1BP3-mediated inhibition of autophagy. We find that endogenous cortactin co-immunoprecipitates with GFP-HS1BP3 (Supplementary Fig. 3a). The interaction is mediated by the SH3 domains of cortactin and HS1 (Supplementary Fig. 3b, lane 9 and 11), and is lost when a critical SH3 domain tryptophan is mutated to tyrosine (Supplementary Fig. 3b, lane 10 and 12). The HS1 and cortactin SH3 domains interact exclusively with the C-terminal part of HS1BP3 (HS1BP3-ΔPX, Supplementary Fig. 3c) containing four proline-rich regions (Fig. 4a). We can however not detect a role for cortactin in basal or starvation-induced autophagy (Supplementary Fig. 3d–e), indicating that binding of HS1BP3 to cortactin is not essential for its inhibitory function in autophagy.

**HS1BP3 localizes to ATG9–ATG16L1-positive membranes**. To identify the mechanisms underlying the role of HS1BP3 as a negative regulator of autophagy, the localization of HS1BP3 to autophagy-related membranes in HEK293 and U2OS cells was investigated. Cells were transfected with GFP- or mCherry-tagged HS1BP3 and their co-localization with WIPI2, ATG9, ATG16L1 or LC3 analysed by confocal imaging. While HS1BP3 is only occasionally detected on WIPI2-positive structures (Fig. 3a, white arrows), it co-localizes well with ATG9 and ATG16L1-positive membranes (Fig. 3a,b). Endogenous HS1BP3 also clearly co-localizes with endogenous ATG9 (Fig. 3c) and with GFP-ATG16L1-positive vesicles (Fig. 3d), and the co-localization is lost after HS1BP3 depletion (Fig. 3d, white arrowheads), demonstrating the specificity of the HS1BP3 antibody. In contrast, HS1BP3 does not show much co-localization with LC3-positive structures (Fig. 3e; Supplementary Fig. 2a) and does not interact with LC3 or GABARAP proteins (Supplementary Fig. 2b). Moreover, no co-localization of endogenous HS1BP3 with GFP-p62, -DFCP1 or -ATG14 is detected (Supplementary Fig. 2a).

Trafficking of ATG16L1 and ATG9 through recycling endosomes is important for autophagosome biogenesis[13–16]. Using confocal and live cell imaging, we find that the HS1BP3-, ATG9- and ATG16L1-positive membranes contain transferrin and transferrin receptor (TfR; Fig. 3b; Supplementary Fig. 5c) and seem to fuse with LC3-positive structures (Supplementary Movies 1–3), indicating they are recycling endosome-derived membranes. Depletion of HS1BP3 does not cause any quantitative changes in the early phagophore/omegasome markers GFP-DFCP1, WIPI2 and ATG16L1. Neither the total intensity of GFP-DFCP1 spots (Supplementary Fig. 2c) nor the number of endogenous WIPI2 or ATG16L1 spots (Supplementary Fig. 2d–e) are affected by HS1BP3 depletion. Taken together, we find that HS1BP3 localizes to ATG9 and ATG16L1-positive recycling endosome-derived membranes that contribute membrane to the forming autophagosome at a stage after omegasome formation, indicating that HS1BP3 regulates autophagy downstream or in parallel of the initial phagophore nucleation step.

**HS1BP3 binds PA and regulates cellular PA levels**. To investigate a possible role for HS1BP3 in regulation of membrane trafficking and/or biogenesis at the expanding phagophore, we first set out to characterize the lipid-binding specificity of the HS1BP3 N-terminal PX domain (Fig. 4a). PX domain proteins are known to mainly bind PI3P, but also other phosphoinositide-binding preferences have been described[38,39]. Using lipid-coated membrane strips, we find the HS1BP3 PX domain to bind strongly to PA, whereas full-length HS1BP3 binds PA, as well as monophosphorylated phosphoinositides (Fig. 4b; Supplementary Fig. 4a). The lipid specificities of purified HS1BP3 proteins (full length or PX domain) was further explored using liposome floatation experiments. We find that the binding of both HS1BP3 and the PX domain to PA increase with increasing concentrations of liposome PA (Fig. 4c).

Using liposomes containing various phosphoinositide species, we could confirm that the PX domain of HS1BP3 also has

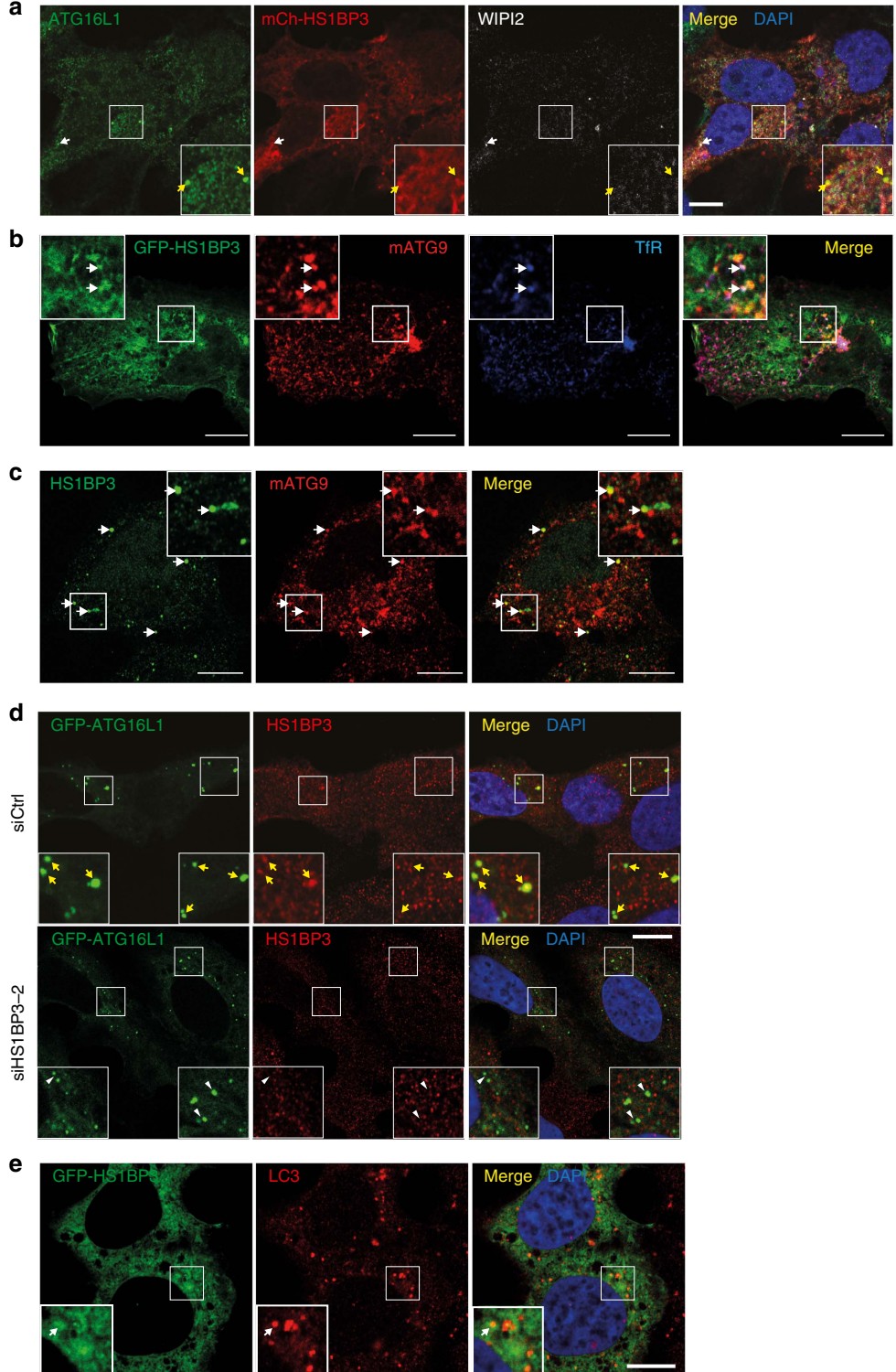

**Figure 3 | HS1BP3 localizes to ATG16L1- and ATG9-positive vesicles.** HEK293 and U2OS cells expressing the indicated proteins were starved for 2 h before fixation and immunostaining with the indicated antibodies. Confocal micrographs show: (**a**) HEK cells expressing mCherry-HS1BP3 stained for endogenous ATG16L1 and WIPI2. Yellow arrows mark HS1BP3- and ATG16L1-positive structures. White arrow marks HS1BP3-, ATG16L1- and WIPI2-positive structure. (**b**) Co-localization of GFP-HS1BP3 with endogenous ATG9 and TfR (white arrows show triple co-localization) in U2OS cells. (**c**) Co-localization of endogenous HS1BP3 with endogenous ATG9 in HEK cells. (**d**) Control or HS1BP3-depleted U2OS cells expressing GFP-ATG16L1 stained for endogenous HS1BP3. Yellow arrows indicate ATG16L1-positive structures that are positive for HS1PB3, while white arrow heads indicate ATG16L1-positive structures that are not positive for HS1BP3. Note that in addition to the specific staining (co-localization with ATG16L1), the HS1BP3 antibody also recognizes other proteins non-specifically both on immunofluorescence and western blotting (Fig. 1d). (**e**) HEK cells expressing GFP-HS1BP3 stained for endogenous LC3. Scale bars, 10 μm.

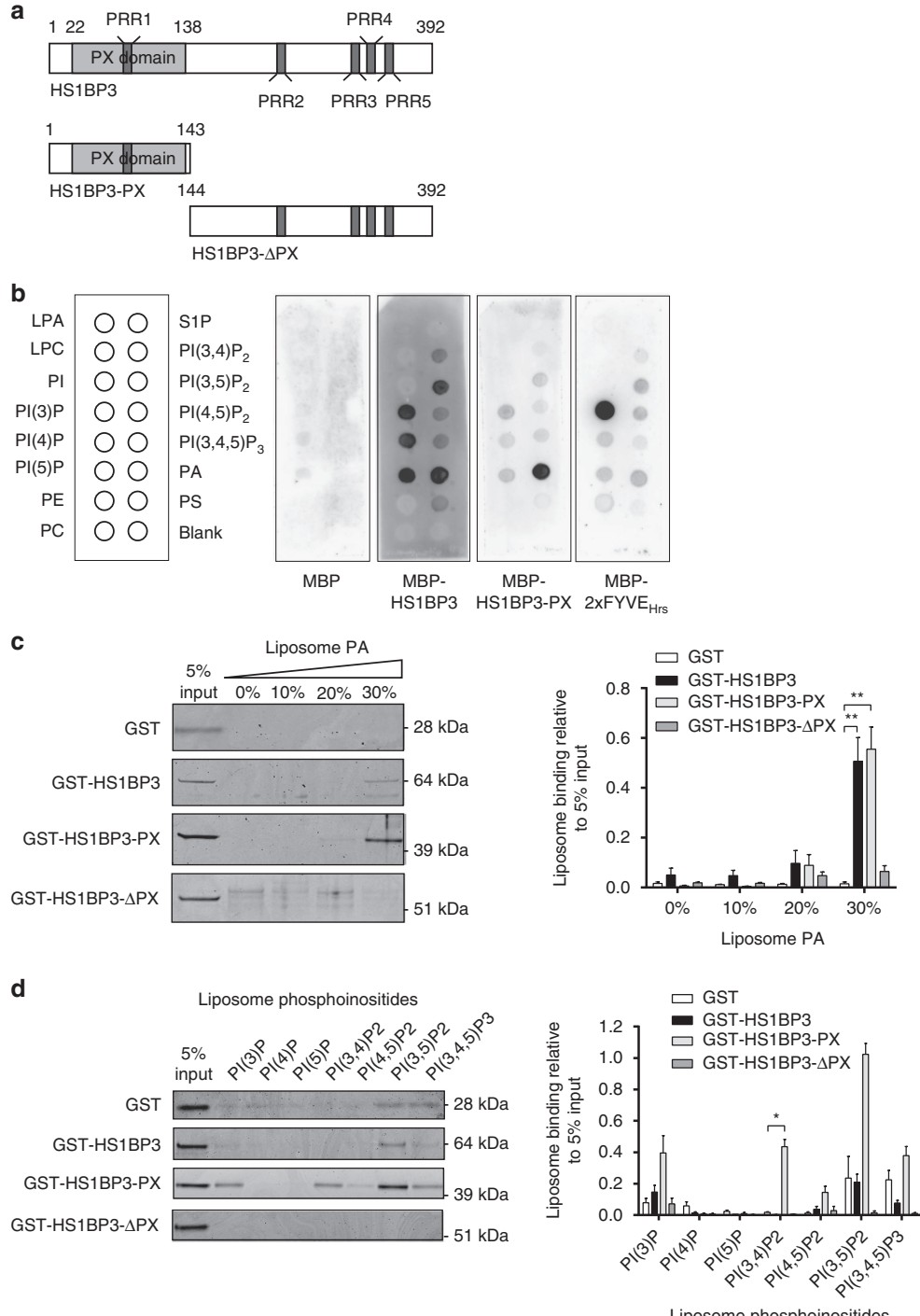

**Figure 4 | HS1BP3 binds PA through its PX domain. (a)** Domain structure of HS1BP3: an N-terminal PX domain followed by an unstructured C terminus. The positions of five proline-rich regions (PRRs) are indicated. HS1BP3 truncations lacking the C-terminal (HS1BP3-PX) or the PX domain (HS1BP3-ΔPX) are shown. **(b)** Membranes spotted with the indicated lipids were incubated with 1 μg ml$^{-1}$ of the indicated recombinant MBP-tagged proteins in a lipid protein overlay assay and bound proteins were detected with anti-MBP immunoblotting. **(c,d)** Liposomes with the indicated molar ratios of dioleyoyl-phosphatidic acid (DOPA) **(c)** or the indicated phosphoinositides **(d)** were incubated with GST or GST-tagged HS1BP3 protein constructs. Protein binding to liposomes was analysed by a lipid floatation assay. Representative coomassie-stained gels are shown and quantified from three independent experiments (mean ± s.e.m.). Significance is calculated as compared with GST control. If significance is calculated as compared to GST-HS1BP3-ΔPX, then GST-HS1BP3-PX shows significantly increased binding to PI(3)P, PI(3,4)P2, PI(4,5)P2, PI(3,5)P2 and PI(3,4,5)P3. *$P < 0.05$, **$P < 0.01$, by Student's $t$-test.

increased affinity for PI(3)P, PI(3,4)P2 and PI(3,5)P2 as compared with the GST control (Fig. 4d). The full-length HS1BP3 protein also shows affinity for phosphoinositides and if compared with GST-HS1BP3-ΔPX it binds several of the same phosphoinositides as the PX domain alone, most notably PI(3)P and PI(3,5)P2 (Fig. 4d). The discrepancy between the data obtained using lipid strips and liposomes is most likely due to differences in the context that the lipids are presented,

where the liposomes more closely resemble the context proteins bind to membranes in cells. We conclude that HS1BP3 binds to PA and several phosphoinositides and we speculate that HS1BP3 has phosphoinositide affinity *in vivo* for PI(3,5)P2 and PI(3)P. Interestingly, the HS1BP3 PX domain co-localizes with ATG16L1 (Supplementary Fig. 4b), suggesting that the lipid affinity of the PX domain contributes to its specific recruitment to ATG16L1-positive membranes.

To get an unbiased quantification of a wide range of lipids in HS1BP3-depleted cells compared with control cells under starvation conditions, the total cellular lipid content was measured by lipidomics (Fig. 5a). Interestingly, whereas the quantities of several lipids are significantly altered (Fig. 5a; Supplementary Fig. 4c); the quantitatively biggest effect is on PA levels, as depletion of HS1BP3 causes a twofold increase in the total abundance of the lipid (Fig. 5a). In total, the levels of 9 out of

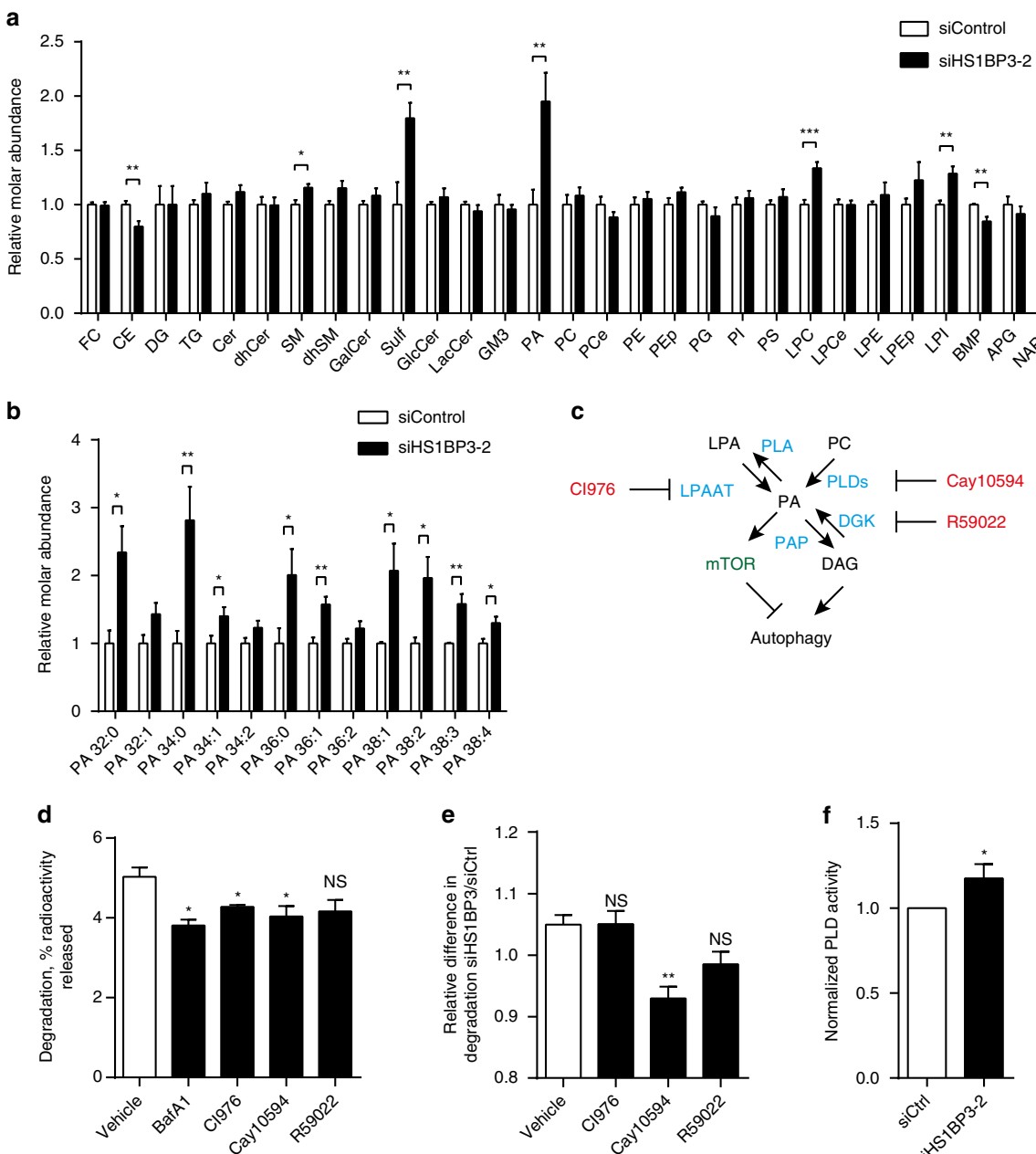

**Figure 5 | Autophagy is dependent on PA synthesis and HS1BP3 affects PLD activity.** (**a**) HEK cells were treated with non-targeting or HS1BP3 siRNA and starved for 2 h. The total lipid content was extracted and analysed by mass spectrometry. Only lipid species that were present in all experimental runs were included. The measured lipid concentrations were first normalized to total lipid per sample to determine molar percentages for each lipid subclass and species, before normalizing to the average molar percentage of controls (mean ± s.e.m., $n = 6$). For a full list of all lipids with abbreviations, see Supplementary Table 1. The lipidomics data sets have been deposited in the Dryad Digital Repository (doi:10.5061/dryad.gq3fk). (**b**) The relative molar abundance of all 12 PA species is shown. (mean ± s.e.m., $n = 6$). (**c**) Schematic overview of the enzymes (blue) governing the generation and turnover of PA from other lipid species (black) and the drugs (red) used to inhibit these pathways. (**d**) Degradation of long-lived proteins in HEK cells was quantified as the release of $^{14}C$-valine after 4 h starvation in the presence of the indicated inhibitors (mean ± s.e.m., $n = 3$). (**e**) The relative difference in long-lived protein degradation between HEK cells with siCtrl and siHS1BP3 was measured for the indicated inhibitors (mean ± s.e.m., $n = 3$). (**f**) PLD activity was measured in HEK lysates of cells transfected with non-targeting or HS1BP3 siRNA (mean ± s.e.m., $n = 3$). $*P < 0.05$, $**P < 0.01$, by Student's $t$-test.

12 PA species are increased by HS1BP3 depletion (Fig. 5b). Interestingly, many of these PA species have been shown to be products of PLD activity[40,41].

PA is a cone-shaped lipid that has been found to stimulate both autophagosome biogenesis[21,22] and autophagosome–lysosome fusion[23]. On the other hand, PA has been shown to activate mTOR signalling[19], which inhibits autophagy. We therefore asked whether depletion of HS1BP3 might stimulate autophagy through changes in mTOR activity. Phosphorylation of the mTORC1 substrate S6 Kinase (pS6K) is however not affected by HS1BP3 depletion (Supplementary Fig. 4d), indicating that the increased PA levels seen on HS1BP3 depletion is not inducing autophagy through decreased mTORC1 signalling.

**HS1BP3 regulates PA levels and autophagy through PLD1**. Several metabolic pathways lead to PA formation (Fig. 5c)[42]. To first determine which of these contribute to increased autophagic flux, cells were treated with inhibitors specific for each pathway followed by quantification of autophagic degradation of long-lived proteins (Fig. 5d). We find that inhibitors of PLD activity (Cay10594, inhibits both PLD1 and PLD2) and lysophosphatidic acid acyltransferases (LPAATs; CI-976) inhibit autophagic flux, whereas an inhibitor of diacylglycerol kinase (R50922) has no significant effect (Fig. 5d). These observations taken together with the increased levels of PA and autophagy on HS1BP3 depletion suggested that HS1BP3 may be inhibiting autophagy as a negative regulator of one of these PA-generating pathways. We reasoned that if HS1BP3 depletion increased the activity of one of the pathways, then chemical inhibition of this pathway should abolish the increased autophagy seen in HS1BP3-depleted cells. Indeed, the relative inhibition of autophagy by the PLD inhibitor Cay10594 is significantly larger in cells depleted of HS1BP3, compared with control cells and cells treated with inhibitors of LPAATs or diacylglycerol kinase (Fig. 5e), suggesting that HS1BP3 may regulate PA levels through PLDs. In line with this, we find that the total activity of PLD enzymes is increased in cell lysates of HS1BP3-depleted cells compared with control cells (Fig. 5f), in line with our data showing that HS1BP3 depletion leads to increased PA levels (Fig. 5a).

To determine which of the PLD enzymes are contributing to the HS1BP3 phenotype, we analysed the localization of GFP-PLD1 and -PLD2 in relation to the autophagy markers LC3 and ATG16L1. GFP-PLD2 staining is concentrated at the plasma membrane with no co-localization with either ATG16L1 (Fig. 6a) or LC3 (Supplementary Fig. 5a). In contrast, GFP-PLD1 shows extensive co-localization with ATG16L1-positive puncta (Fig. 6a), and is often seen in close proximity to LC3-positive puncta (Supplementary Fig. 5a,c). The majority of the vesicles stained by GFP-PLD1 and ATG16L1 contain transferrin and TfR (Fig. 6b), and seem to fuse with Cherry-LC3B-positive structures (Supplementary Fig. 5c; Supplementary Movies 1 and 2), indicating that they are recycling endosome-derived vesicles destined for autophagosome biogenesis. HS1BP3 is also detected at the TfR- and PLD1-positive structures (Supplementary Fig. 5b). Because PLD1, and not PLD2, localize to ATG16L1-positive membranes (Fig. 6a), we conclude that the effect of HS1BP3 on PLD activity is most likely through the regulation of PLD1 on recycling endosomes or vesicles derived thereof.

To get some mechanistic insight into how HS1BP3 regulates PLD1 activity at the Atg16L1-positive membranes, we investigated the localization of PLD1 to ATG16L1-positive structures in the absence or presence of HS1BP3. Interestingly, there is a significant increase in the amount of ATG16L1-positive vesicles with GFP-PLD1 staining in HS1BP3-depleted cells compared

with control cells (Fig. 6c), indicating that HS1BP3 may regulate PLD1's access to these vesicles.

As both HS1BP3 and PLD1 have a PX domain with similar lipid-binding specificities (Fig. 4; ref. 43), we hypothesized that they might compete for binding to lipids in ATG16L1-positive membranes. In line with this notion, the co-localization between HA-PLD1 and endogenous ATG16L1 is significantly decreased in cells expressing the full length or PX domain of GFP-HS1BP3 compared with control cells expressing GFP only, but not in cells expressing HS1BP3 lacking the PX domain (GFP-HS1BP3 ΔPX; Fig. 6d,e).

We further looked for protein–protein interactions between these proteins that could explain their apparent co-localization and functional relationship, but were unable to detect any interactions between over-expressed or endogenous PLD1, HS1BP3 and ATG16L1 under the conditions tested (Supplementary Fig. 6a). Taken together, our data indicate that HS1BP3 prevents access of PLD1 to ATG16L1-positive vesicles and we speculate that the two proteins compete for binding to lipids in the ATG16L1-positive membranes.

To further map the functional relationship between HS1BP3 and PLD1 in regulation of autophagy, endogenous LC3 puncta were quantified in cells depleted of HS1BP3 and/or PLD1, and at the same time transfected with GFP, GFP-HS1BP3 or GFP-PLD1 (Fig. 7a). While depletion of HS1BP3 increases the number of LC3 spots and LC3-II levels (in line with data in Fig. 1), this effect is reversed in cells co-depleted of PLD1 (Fig. 7a; Supplementary Fig. 6b), indicating that the increase in autophagy seen with HS1BP3 depletion is dependent on PLD1, in line with our previous observation of the HS1BP3-mediated increase in autophagy being reverted by the use of PLD inhibitors (Fig. 5e). Moreover, overexpression of GFP-PLD1 causes a significant increase in the amount of LC3-positive spots (Fig. 7a), with no further increase on concurrent depletion of HS1BP3, further supporting that HS1BP3 and PLD1 are part of a common mechanism, where HS1BP3 acts on autophagy through regulation of PLD1.

Taken together, our observation that HS1BP3 inhibits the localization of PLD1 to ATG16L1-positive vesicles suggests that the effect of HS1BP3 on autophagy is through the regulation of PLD1-generated PA on ATG16L1-positive autophagosome precursors (Fig. 7b). In light of the binding of HS1BP3 to PA, we propose a mechanism by which HS1BP3 regulates PLD1 by providing negative feedback on PA production through regulating the access of PLD1 to target membranes.

## Discussion

Autophagosome formation must be properly regulated to balance the need to ensure cellular quality control and nutrient availability during starvation, with the necessity to prevent excessive autophagosome formation and detrimental degradation of crucial cellular components. We here identify the PX domain protein HS1BP3, as an inhibitor of the autophagosome formation and autophagic degradation through a negative-feedback mechanism, involving the regulation of PLD1 activity, and hence PA levels, in ATG16L1-positive membranes. Depletion of HS1BP3 affects both PLD activity in lysates and PLD1 localization to ATG16L1-positive vesicles, suggesting that HS1BP3 acts as a sensor and regulator of local PA levels. On its recruitment, HS1BP3 will inhibit PLD1 activity on ATG16L1-positive autophagosome precursors, thereby reducing their PA content and autophagosome formation (Fig. 7b).

Many PX domains bind PI(3)P, but there are also other demonstrated lipid-binding specificities[38,39]. We found that both the full-length protein and the PX domain of HS1BP3 bind to

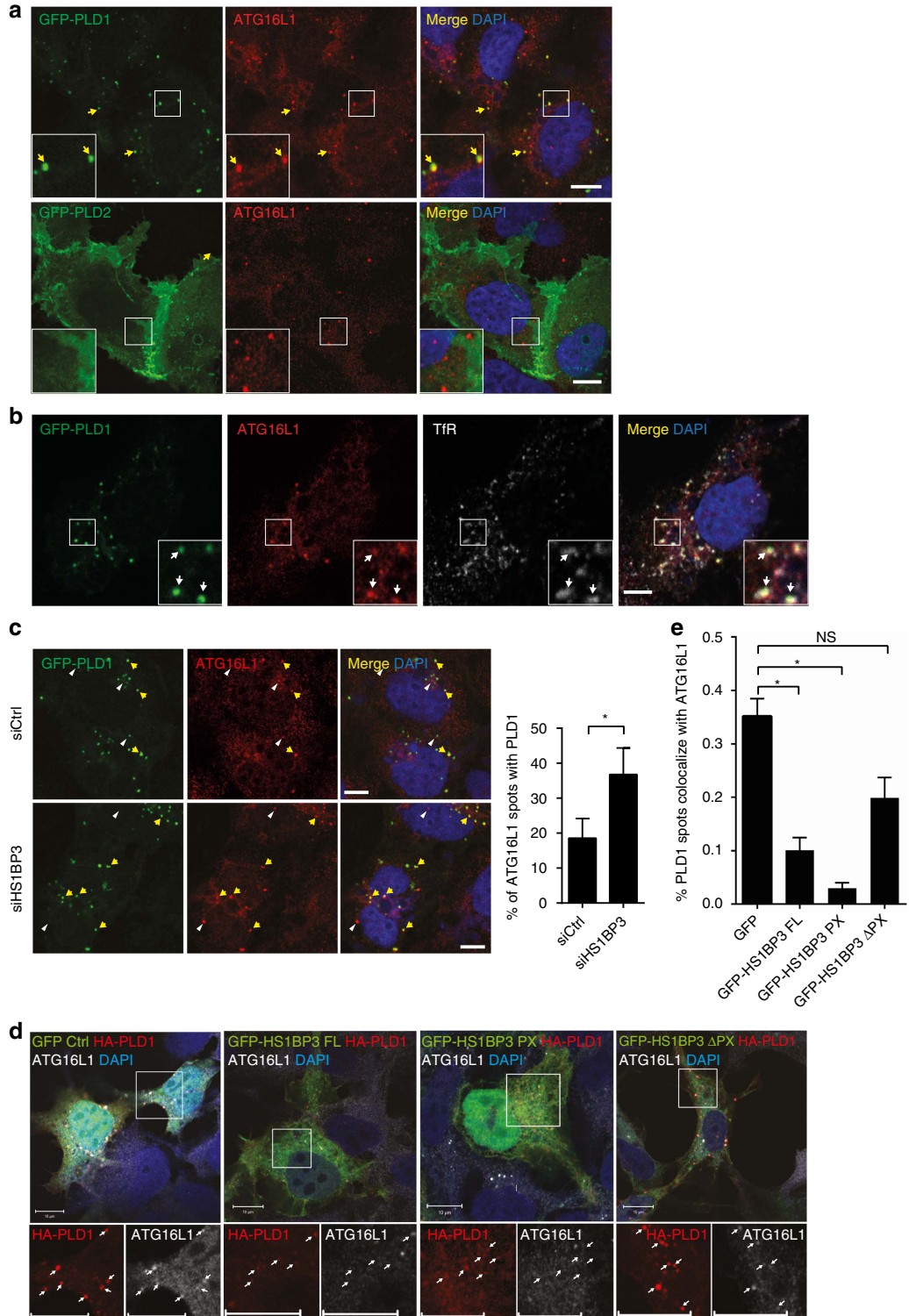

**Figure 6 | PLD1 co-localization with ATG16L1 is affected by HS1BP3.** (**a**) HEK cells were transfected with GFP-tagged PLD1 or PLD2. After starvation and fixation the cells were immunostained for ATG16L1 and analysed by confocal microscopy. (**b**) HEK cells were transfected with GFP-PLD1, starved, fixed and immunostained for ATG16L1 and TfR. (**c**) HEK cells were first treated with non-targeting or HS1BP3 siRNA, then transfected to express GFP-PLD1, starved, fixed and immunostained for ATG16L1. Yellow arrows indicate ATG16L1 vesicles positive for PLD1 and white arrow heads indicate ATG16L1 vesicles negative for PLD1. Co-localization of GFP-PLD1 to ATG16L1 vesicles was quantified in transfected cells using the ImageJ plugin Squassh, using 10 pictures of each condition from three independent experiments (mean ± s.e.m., $n = 3$). (**d**) HEK cells were transfected with HA-PLD1 together with GFP, GFP-HS1BP3 full-length, -PX or ΔPX constructs, starved and stained for endogenous ATG16L1. Arrows indicate co-localization between ATG16L1 and HA-PLD1.
(**e**) Co-localization of HA-PLD1 with endogenous ATG16L1 vesicles was quantified in transfected cells in **d** with the Zen software (Zeiss) using 10 pictures of each condition from three independent experiments (mean ± s.e.m., $n = 3$). Scale bars, 10 μm. *$P < 0.05$, by Student's $t$-test.

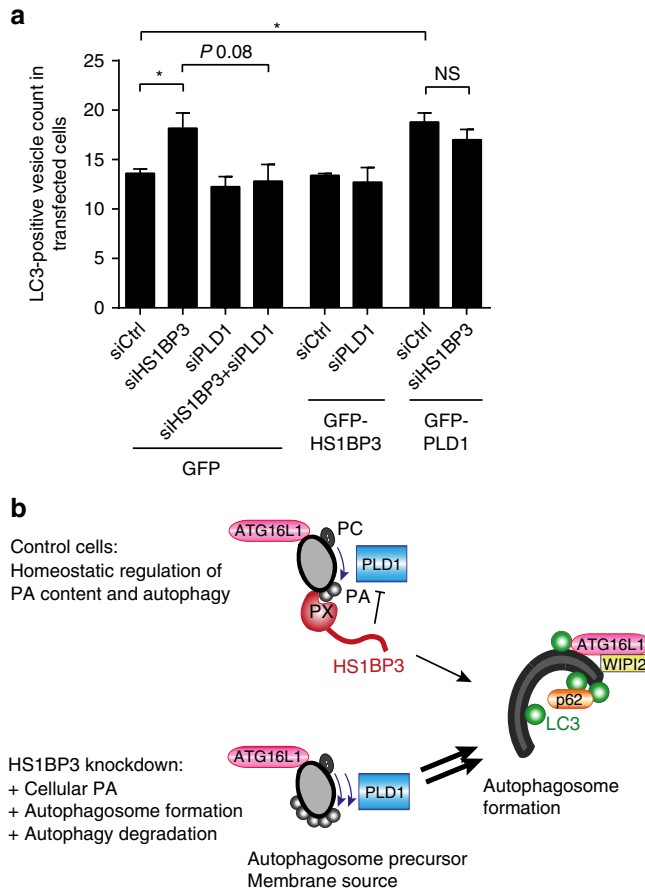

**Figure 7 | HS1BP3 regulates autophagy through PLD1. (a)** HEK cells were first treated with the indicated siRNA and then transfected with the indicated GFP-tagged construct. Cells were starved and fixed before immunostaining for endogenous LC3. LC3 spots were counted only in transfected cells, minimum 200 transfected cells per condition in three independent experiments (mean ± s.e.m., $n = 3$). *$P < 0.05$, by Student's $t$-test. **(b)** Model for the role of HS1BP3 in autophagy. PLD1 generates PA on ATG16L1-positive autophagosome precursor membranes. HS1BP3 is recruited to these membranes by the generated PA, inhibiting PLD1 activity and displacing it from the ATG16L1 vesicles. HS1BP3 thus provides a negative feedback on PA generation on these vesicles. If HS1BP3 is depleted from the cells, this negative feedback is lost, causing the PA concentrations of these membranes to increase and thereby drive increased autophagosome formation.

PA and phosphoinositides. PA binding by PX domains has previously been reported. A study of the PX domain of PLD1 demonstrated binding to PI(3,4,5)P3, PI(3)P, as well as other PI species and a moderate affinity for PA in a separate binding pocket on the PX domain[43]. Interestingly, the simultaneous binding of both sites was shown to increase the membrane affinity of the PX domain[43]. Similarly, the PX domain of the p47 subunit of NADPH oxidase was found to simultaneously bind PI(3,4)P2 and PA in separate binding pockets, increasing its membrane affinity[44]. Strikingly, we observed a competition between HS1BP3 and PLD1 for binding to ATG16L1-positive precursors. Having a similar lipid-binding specificity may be the basis of this competition between HS1BP3 and PLD1 for membrane binding, and this is something that will be interesting to explore further in future studies.

We also found that the LPAAT pathway of PA generation contributes to autophagy, demonstrating that the involvement of PA in autophagy is not limited to PLD1. The exact mechanism(s)

underlying the role of PA in stimulation of autophagosome biogenesis is not clear and might be related to its role as a second messenger, but could also be associated with PA having a direct structural role in membrane curvature and/or fusion due to the unique characteristics of PA in a lipid bilayer. PA is the only anionic phospholipid that induces negative membrane curvature due to its cone shape under physiological conditions[45]. Cone-shaped lipids such as PA also facilitate penetration of proteins into the membrane, since the lipid head groups are more loosely packed[46], as shown to be important for insertion of ATG3 in the forming phagophore and subsequent lipidation of LC3/GABARAP[47].

Another relevant characteristic of PA is the demonstrated fusogenic properties of this lipid. PLD activity has been demonstrated as essential for various vesicle fusion events, such as sporulation in yeast[48], mitochondrial fusion[49] and exocytosis[50]. We speculate that the increased autophagy seen in HS1BP3-depleted cells might be facilitated by PA-mediated changes in the fusogenic properties of the ATG16L1-positive autophagosome precursors. Homotypic fusion of ATG16L1-positive vesicles has previously been found to facilitate their contribution to autophagosome formation and this was demonstrated to be dependent on the SNARE protein VAMP-7 (ref. 51). Intriguingly, VAMP-7 was recently described as an effector of PLD1 in neurite outgrowth[52], suggesting a possible mechanism by which PLD1-generated PA could affect autophagy. HS1BP3 is detected on ATG16L1 vesicles that also contain ATG9 and TfR, suggesting they are of recycling endosome origin. We recently identified the PX-BAR protein SNX18 as a positive regulator of autophagosome biogenesis by tubulation of recycling endosome membrane for the delivery to phagophore nucleation sites[13]. The RAB11-binding protein TBC1D14 is another negative regulator of autophagy found to regulate the recycling endosome membrane remodelling[15]. It will be interesting to explore how these membrane-associated proteins regulate the recycling endosome dynamics and how potential qualitative changes in these vesicles affect autophagy through homotypic and possibly heterotypic fusion processes.

In conclusion, we have identified HS1BP3 as a novel negative regulator of autophagy and cellular PA levels. We propose that PLD1 generates PA on ATG16L1-positive autophagosome precursor membranes and that HS1BP3 is recruited to these membranes by binding to PA (Fig. 7b). HS1BP3 affects the ability of PLD1 to generate PA, as well as regulating the access of PLD1 to ATG16L1-positive vesicles, changing the properties of these membranes and thereby providing a homoeostatic regulation of autophagy. HS1BP3 thus functions as a negative-feedback mediator of PA levels to regulate autophagosome formation.

## Methods

**Cell lines and inhibitors.** HeLa, HEK and U2OS cells were from American Type Culture Collection and were maintained in Dulbecco's modified Eagle's medium (Gibco) supplemented with 10% fetal bovine serum (FBS), 5 U ml$^{-1}$ penicillin and 50 µg ml$^{-1}$ streptomycin. The HEK 293A GFP-LC3 cell line[53] was a kind gift from S. Tooze, Cancer Research UK, London, UK. The HEK GFP-DFCP1 cell line[6] was a kind gift from N. Ktistakis, Babraham Institute, Cambridge, UK. The HEK GFP-p62 cell line was a kind gift from G. Bjørkøy, HiST, Trondheim, Norway. All cell lines have been tested negative for mycoplasma. Bafilomycin A1 (Enzo Lifesciences) was used at 100 nM. CI 976 (Tocris Bioscience), Cay10594 (Cayman Chemical) and R59022 (Tocris Bioscience) were used at 20 µM. Glass support was coated by 20 µg ml$^{-1}$ fibronectin (Sigma) before plating HEK cell lines to avoid the cells from detaching from the surface. For starvation in nutrient-deplete medium, the cells were incubated in Earls Balanced Salt Solution (EBSS; Invitrogen), with the exception of the HEK GFP-DFCP1 cells that were starved as described previously[6] in 140 mM NaCl, 1 mM CaCl$_2$, 1 mM MgCl$_2$, 5 mM glucose and 20 mM Hepes, pH 7.4.

**Antibodies and dyes.** The following primary antibodies were used: mouse anti-cortactin (Upstate, 05-180, 1:1,000), mouse anti-GFP (Clontech, 632381,

1:1,000), mouse anti-Flag (Sigma, F1804, 1:500), mouse anti-MBP (NEB, e8032S, 1:10,000), rabbit anti-ULK1 (Santa Cruz, sc-33182, 1:250), rabbit anti-HS1BP3 (GeneTex, GTX107715, 1:10,000 for WB and 1:500 for IF), rabbit anti-LC3 (Cell Signaling, 27755, 1:1,000 for WB), mouse anti-β-actin (Sigma, SAB1305567 1:20,000), mouse anti-myc (DSHB, 9E10, 1:20), mouse anti-alpha tubulin (Sigma, T5168, 1:20,000), rabbit anti-LC3 (MBL, PM036, 1:500 for IF), mouse anti-p62 (BD biosciences, 610833, 1:1,000 for WB), goat horseradish peroxidase (HRP)-conjugated anti-GST (Abcam, ab58626, 1:10,000 for phosphatidylinositol phosphate (PIP) strips), rabbit anti-phospho-AKT Ser473 (Cell Signaling, 4060, 1:2,000), rabbit anti-phospho-p70-S6K Thr389 (Cell Signaling, 9202, 1:1,000), rabbit anti-p70-S6K (Cell Signaling, 9205, 1:1,000), rabbit anti-ATG16L1 (MBL, PM040, 1:200), mouse anti-TfR CD71 (Santa Cruz, sc-65877, 1:200), mouse anti-WIPI2 (kind gift from Sharon Tooze, 1:2,000), rabbit anti-PLD1 (Cell Signaling, 3832S, 1:200), mouse anti-HA (Abcam, ab18181, 1:200), hamster anti-mAtg9 (kind gift from Sharon Tooze, 1:1,000). HRP- and Cy2/3/5-conjugated secondary antibodies were obtained from Jackson Immunolabs. Far-red fluorophore-conjugated secondary antibodies were from LI-COR. Transferrin–Alexa 647 was from Invitrogen.

**Transfection of siRNA oligonucleotides and western blotting.** siRNA oligonucleotides were Dharmacon ON-TARGET plus; HS1BP3-1J-013029-09 AAGAAGGAGUGACCGGUAU, HS1BP3-2J-013029-10 UGAAGAGGCUUUCG ACUUU, HS1BP3-3J-013029-11 GAGCCUGAAGGGCGAGGAU, HS1BP3-4J-013029-12 UCCCAAAGUGGCCGUGAAA, ULK1 J-005049-06 CCACGCAG-GUGCAAGACUA, cortactin CCCAAGAACACUAUGUGAAAGGG[54] and PLD1 SmartPool consisting of four oligonucleotides pooled together. An amount of 20–100 nM siRNA was delivered to the cells by Lipofectamine RNAi max (Invitrogen). To demonstrate specific protein knockdown and monitor LC3 levels, the cells were lysed in 25 mM Hepes pH 7.5, 125 mM K-acetate, 2.5 mM Mg-acetate, 5 mM EGTA, 1 mM DTT and 0.5% NP-40 supplemented with Complete protease inhibitor (Roche). Protein concentration was measured by Biorad Protein Assay to run equal amounts of cell lysate on SDS–polyacrylamide gel electrophoresis (PAGE), followed by western blotting using specified primary antibodies and secondary antibodies for enhanced chemiluminescent (ECL) detection or far-red fluorescence. For ECL detection: membranes were incubated with HRP-conjugated secondary antibodies detected by the Supersignal West Dura Extended Duration Substrate kit (Pierce). Imaging and quantification of protein levels were performed using the Syngene gel documentation unit, Genesnap acquisition software and GeneTools analysis software. For far-red fluorophores: membranes were incubated with far-red fluorophore-conjugated secondary antibodies, and detection and analysis was performed by LI-COR Odyssey imaging. Uncropped scans of western blots are found in Supplementary Fig. 7.

**Plasmids and transfection for ectopic expression.** HS1BP3 complementary DNA (cDNA) was amplified by PCR using primers 5′-ATAGTCGA-CATGCAGTCCCCGGCGGTGCTC-3′ and 5′-ATAGCGGCCGCTCAGAAGA-GACTGGGGGCGG-3′ from a cDNA library made by reverse transcription (Biorad iScript) from mRNA isolated from HEK cells, TA cloned into pCR2.1-TOPO (Invitrogen) and subcloned into pENTR1A (Invitrogen) using SalI and NotI restriction sites. From there, sequences coding for HS1BP3-PX and HS1BP3-ΔPX were amplified by PCR using primers 5′-ATAGTCGACATGCAGTCCCCGGCGG TGCTC-3′and 5′-ATAGCGGCCGCTCAGGATCTGGTACCTAAGAACTC-3′ or 5′-ATAGTCGACGCTGCAGGGCTCACCAGCAG-3′ and 5′-ATAGCGGCCGCT CAGAAGAGACTGGGGGCGG-3′, respectively, and cloned into pENTR1A (Invitrogen). Tagged variants were made by Gateway LR cloning (Invitrogen) into respective pDEST vectors (Invitrogen). See Supplementary Table 2 for a full list of plasmids used in this study. To transfect cells with plasmids encoding GFP- or mCherry-tagged HS1BP3 variants, the plasmids were delivered to the cells by forward transfection with FuGene (Roche) or Lipofectamine 2000 (Invitrogen) before further treatment as described.

**Microscopy.** siRNA-treated HEK GFP-LC3 cells grown on glass support, were starved or not for 2 h, pre-permeabilized on ice with 0.05% saponin in 80 mM K-Pipes pH 6.8, 5 mM EGTA and 1 mM MgCl₂ before fixation in 3% paraformaldehyde (PFA) or fixed directly in methanol for 10 min at −20 °C. The nuclei were counterstained with 1 µg ml⁻¹ Hoechst in phosphate-buffered saline or mowiol. The number of GFP-LC3 spots was quantified using the automated Olympus ScanR microscope equipped with a ULSAPO 40 × objective and the corresponding analysis program or by an automated Zeiss CellObserver equipped with a 40 × EC Plan Neofluar objective, and using the physiology module of the Zeiss Assaybuilder software. For immunostaining and confocal analysis, cells were grown on glass cover slips and after the described treatments, fixed in 3% PFA for 15 min on ice or in methanol for 10 min at −20 °C and mounted in mowiol containing 1 µg ml⁻¹ Hoechst or 4,6-diamidino-2-phenylindole to stain the nuclei. Confocal images were taken on an Olympus confocal microscope equipped with a UPlanSApo 60 × objective. Co-localization of stainings of interest from confocal images was quantified using the ImageJ-based Squassh plugin[55]. The plugin subtracts background, segments the image into vesicles in each channel and co-localization is quantified as degree of signal intensity overlap in the segmented

regions. Cell mask thresholding was used to include vesicles only in transfected cells.

For live cell imaging, HEK293A cells were plated in complete media in wells of Lab-Tek II chambered coverglass (2 × 10⁴ cells per well), precoated with poly-D-lysine and transfected with indicated constructs. Complete media was removed 24 h after transfection, cells were washed 2 × in phosphate-buffered saline, then starved for 1 h in EBSS containing 5 µg ml⁻¹ of transferrin–Alexa Fluor 647 conjugate and imaged live with Zeiss LSM710 confocal microscope (63 × 1.4 plan-apochromat objective, single plane).

**Quantitative PCR.** siRNA-transfected cells were frozen dry at −80 °C, RNA isolated by RNeasy plus kit (Qiagen), cDNA synthesized by reverse transcription (Biorad iScript) and quantitative real-time PCR performed using SYBRGreen (Qiagen), and pre-designed Quantitect (Qiagen) primer sets for the described targets relative to SDHA or TBP as housekeeping genes on a Lightcycler 480 (Roche Applied Science) or on a CFX96 (Bio-Rad).

**GFP-p62 measured by flow cytometry.** The GFP-p62 flow cytometry assay was described previously[32]. Briefly, HEK GFP-p62 cells in 24-well plates were transfected with siRNA and 24 h later induced with 1 ng ml⁻¹ doxycyclin to express GFP-p62 for 48 h. GFP-p62 expression was shut off and the cells were starved in EBSS for 2.5 h or treated as indicated. The cells were then trypsinized and passed through cell strainer caps (BD Biosciences) to obtain single-cell suspensions. Cells were analysed on a FACSAria cell sorter running FACSDiva software version 5.0 (BD Biosciences) using the blue laser for excitation of GFP. GFP fluorescence was collected through a 530/30 nm band-pass filter in the E detector. Data were collected from a minimum of 10,000 singlet events per tube, and the median GFP-p62 value was used for quantification.

**Long-lived protein degradation.** To measure the degradation of long-lived proteins by autophagy, proteins were first labelled with 0.25 µCi ml⁻¹ L-¹⁴C-valine (Perkin Elmer) for 24 h in GIBCO-RPMI 1640 medium (Invitrogen) containing 10% FBS. The cells were washed and then chased for 3 h in nonradioactive Dulbecco's modified Eagle's medium (Invitrogen) containing 10% FBS and 10 mM valine (Sigma), to allow degradation of short-lived proteins. The cells were washed twice with EBSS (Invitrogen), and starved or not for 4 h in the presence or absence of 10 mM 3-methyladenine (Sigma). 10% Trichloroacetic acid was added to the cells before incubation at 4 °C to precipitate radioactive proteins. Ultima Gold LSC cocktail (Perkin Elmer) was added to the samples and protein degradation was determined by measuring the ratio of trichloroacetic acid-soluble radioactivity relative to the total radioactivity detected by a liquid scintillation analyser (Tri-Carb 3100TR, Perkin Elmer), counting 3 min per sample.

**In vitro interaction pull-down assays and immunoprecipitation.** Recombinant GST- or MBP-tagged proteins were expressed and purified from Escherichia coli. 1 µg of proteins of interest were incubated together in NETN buffer (50 mM Tris pH 8, 100 mM NaCl, 6 mM EDTA, 6 mM EGTA, 0.5% NP-40, 1 mM DTT, Roche Complete protease inhibitor) followed by GST pulldown by glutathione sepharose (GE Healthcare). For GST pulldown from cell lysate, cells were lysed in 10 mM TrisHCl pH 7.5, 150 mM NaCl, 0,5 mM EDTA, 0,5% NP-40, protease inhibitor (Roche) and phosphatase inhibitor (Sigma), and the cell lysate incubated with recombinant glutathione sepharose-bound GST proteins. The resulting pulldowns were analysed by immunoblotting. For in vitro translation, indicated GFP fusion proteins were in vitro translated in TNT T7-coupled reticulocyte lysate (Promega L4610) in the presence of ³⁵S-methionine (PerkinElmer) and precleared on glutathione–sepharose before incubation in NETN buffer together with glutathione–sepharose-bound recombinant GST-tagged LC3B or GABARAP proteins expressed in and purified from E. coli according to manufacturer's instructions. The resulting pulldowns were separated by SDS–PAGE. The gels were Coomassie blue stained and the in vitro-translated co-purified proteins were detected by autoradiography on a Typhoon phosphorimaging scanner (GE Healthcare). For immunoprecipitation from lysates, GFP, GFP-HS1BP3 or GFP-PLD1 were immunoprecipitated by GFP trap (Chromotek) following the manufacturer's protocol. The resulting immunoprecipitates or pulldowns were separated by SDS–PAGE and analysed by western blotting.

**Lipid-binding assays.** PIP strips or membrane lipid strips (Echelon biosciences) were blocked in 3% fatty acid-free bovine serum albumin (Sigma) in TBS-T (50 mM Tris pH 7.4, 150 mM NaCl, 0.1% Tween) before incubation with 1 µg ml⁻¹ recombinant MBP proteins in TBS-T. After repeated washing in TBS-T, bound protein was immunodetected with chemiluminescence. Liposomes were prepared by mixing different molar ratios of palmitoyl-oleyl-phosphatidylcholine, dioleoyl-phosphatidic acid, dioleoyl-PE–rhodamine and the PIPs in chloroform and drying the lipids to a thin film. Lipids were reconstituted in a buffer containing 20 mM Hepes, pH 7.4, 150 mM NaCl and 1 mM MgCl2 and exposed to seven cycles of flash-freezing in liquid nitrogen and thawing in a 37 °C water bath, before extruding the lipid mixtures through two polycarbonate filters with 200 nm pores a total of 21 times. Lipid binding to liposomes was

analysed by a floatation assay, where 5 μM of GST-tagged protein was incubated with 2 mM total lipid and 1 mM DTT in a buffer composed of 20 mM Hepes pH 7.4, 150 mM NaCl and 1 mM MgCl$_2$ for 20 min at 25 °C. To separate liposomes and bound protein from free protein, this mixture was subjected to a Nycodenz liposome flotation assay as described in ref. 47. Gradients were centrifuged at 48,000 r.p.m. (280,000$g$) in a SW55Ti rotor (Beckman) for 4 h at 4 °C, and the liposomes and bound protein were recovered from the top 80 μl of the gradient. The lipid recovery from the gradient was determined by measuring dioleyoyl-PE–rhodamine fluorescence using a SpectraMax fluorescence spectrometer. Floatation reactions were analysed on a 12% bis-Tris gel (Novex) by SDS–PAGE, where 10% of the total lipid from each floatation reaction was run together with a protein control containing 0.5% of the total protein. The proteins were visualized with Coomassie Blue stain per the manufacturer's instruction (Imperial Protein Stain, Thermo Scientific).

**High-performance liquid chromatography–mass spectrometry.** Lipid extracts were prepared from total cell lysates (after 2 h starvation) using a modified Bligh/Dyer extraction procedure as previously described[56]. Samples were analysed using an Agilent Technologies 6490 Ion Funnel LC/MS Triple Quadrupole system with front end 1260 Infinity HPLC. Phospholipids and sphingolipids were separated by normal-phase high-performance liquid chromatography (HPLC), while neutral lipids were separated using reverse-phase HPLC. For normal-phase analysis, lipids were separated on an Agilent Rx-Sil column (i.d. 2.1 × 100 mm) using a gradient consisting of A: chloroform/methanol/ammonium hydroxide (89.9:10:0.1) and B: chloroform/methanol/water/ammonium hydroxide (55:39:5.9:0.1), starting at 5% B and ramping to 70% B over a 20 min period before returning back to 5% B. Neutral lipids were separated on an Agilent Zorbax XDB-C18 column (i.d. 4.6 × 100 mm), using an isocratic mobile phase chloroform:methanol:0.1 M ammonium acetate (100:100:4) at a flow rate of 300 μl min$^{-1}$. Multiple reaction monitoring transitions were set up for quantitative analysis of different lipid species and their corresponding internal standards as described previously[56]. Lipid levels for each sample were calculated relative to the spiked internal standards and then normalized to the total amount of all lipid species measured and presented as relative mol %. Data are presented as mean mol % for three samples of each condition.

**PLD activity assay.** PLD activity was measured using the Amplex Red Phospholipase D Assay Kit (Invitrogen) according to the manufacturer's instructions. Cell lysates were made of six-well plates using the assay reaction buffer with 1% Triton X-100. For each replicate, 20 μg of lysate was diluted to 50 μl of lysis buffer and added to 50 μl of reaction mixture. Four technical replicates per sample were distributed in a black half-area 96-well plate (Corning). The plate was covered and the reaction proceeded for 30 min at 37 °C before fluorescence was measured.

**Zebrafish work.** Experimental procedures followed the recommendations of the Norwegian Regulation on Animal Experimentation. All experiments were conducted on GFP-LC3 transgenic larvae[57] under 5 dpf. Translation-blocking antisense morpholino oligonucleotides for *Hs1bp3* (5′-TTcTaATACcTC CcTCTcACATTGT-3′) or a scrambled-sequence morpholino (5′-TTGTTAT ACGTCCGTCTGACATTGT-3′) were designed according to the manufacturer's recommendations (Gene Tools, Philomath, OR, USA) and 16.86 ng of either was injected into embryos at the one-cell stage. Capped full-length human wild-type Hs1bp3 mRNA was transcribed from linearized pSP64 Poly (A) Vector (Promega; Supplementary Table 2) using mMessage mMachine (Ambion) and 50–150 pg was coinjected with the *Hs1bp3* morpholino as described[58]. Microscopic visualization, screening and imaging of the fish were performed on a stereomicroscope Leica DFC365FX with a 1.0 × planapo lens. Control morpholino, *Hs1bp3* morpholino and human *Hs1bp3* mRNA injected live embryos were anaesthetized with tricaine at 2 dpf and mounted in low-melting point agarose for imaging with Olympus FV1000 scanning confocal microscope (under a 60 × /1.00 numerical aperture water immersion objective). Injected embryos were treated or not with 10 μM chloroquine at 28 °C for 6 h and then followed by immunoblot analysis and imaging. Embryos at 2 dpf were deyolked and then homogenized in lysis buffer (50 mM Tris-HCl (pH 8), 150 mM NaCl, 5 mM EDTA, 1% NP-40, 0.5% sodium deoxycholate, 0.1% SDS, protease inhibitor cocktail (Roche)). All experiments were replicated at least three times with 7–13 embryos per condition. Embryos were randomly distributed to receive the described treatments. No blinding was used and no animals were excluded from the analysis.

**Statistics.** The $P$ values were derived from two-tailed $t$-test from Excel (Microsoft) for paired samples, and considered statistically significant at $P \leq 0.05$. In some cases, values were log-transformed to obtain a normal distribution.

**Data availability.** The lipidomics datasets have been deposited in the Dryad Digital Respository (DOI: doi:10.5061/dryad.gq3fk).

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

## Acknowledgements

We thank S. Tooze, N. Ktistakis, G. Bjørkøy, A. Thige, S.S. Taylor, W. Eskild, B.-H. Toh, M. Frohman and J.K. Burkhardt for sharing reagents. This research was supported by the Molecular Life Science program of the University of Oslo and the Norwegian Cancer Society. G.D.P. is supported by NIH grant R21 AG045020. T.J.M is supported by NIH grant R01 GM100930. The authors declare no competing financial interests.

## Author contributions

H.K. and A.S. conceived the idea. H.K., P.H. and K.S. designed and conducted the experiments, analysed the data and generated figures. S.W.S. performed the ultra-structural studies, A.H.L. and T.J.M. performed the liposome interaction studies. S.P., S.R.C., V.H.L. and G.T.B. helped perform experiments. B.J.M. designed and performed the zebrafish experiments. R.B.C., B.Z. and G.D.P performed and analysed the lipidomics study. K.L. contributed to the statistical analysis. H.K., P.H. and A.S. wrote the manuscript with input from all co-authors.

## Additional information

**Competing financial interests:** The authors declare no competing financial interests.

