## [Peer Review File · Nature Communications]

Reviewers' Comments:

Reviewer #1 (Remarks to the Author)

The authors have responded comprehensively to the reviews. Proper controls and additional for experiments were included to strengthen the authors' data and provide a strong argument.

Reviewer #2 (Remarks to the Author)

The formation of autophagosomes is a complex process which requires a complex machinery and multiple steps, involve both lipids, proteins and interdependencies between lipids and proteins. Holland et al have addressed the role of a lipid binding protein, HS1BP3, which binds PA. HS1BP3 is known to bind HS1 or cortactin. The authors demonstrate that HS1BP3 is a negative regulator of starvation-induced autophagy, which colocalizes with Atg16L. HS1BP3 is proposed to act by inhibiting PLD1 activity on ATG16L1-positive and TfR-positive membranes. In addition, loss of HS1BP3 increases the synthesis of particular PA species in cells. The authors propose a model whereby HS1BP3 regulates PLD1's access to the Atg16L1, TfR-positive membranes thereby controlling the lipid composition, in particular PA levels of autophagosome precursor membranes. This is a revised version of a manuscript submitted to NCB. The authors have addressed some of the points raised in the previous review but they have done this primarily by removing large sets of data or replacing all the original problematic data. They have made a effort to identify an antibody to the endogenous protein but this has not provided insight into the mechanism. The manuscript, while much easier to understand, is now largely descriptive and provides no definitive mechanism to understand how HS1BP3 works to alter PA levels.

Other major points:

1. What are the ATG16L- TfR- positive membranes? How do the authors believe they contribute to autophagosome formation if they only contain ATG16L1 and no up or downstream Atg proteins.
2. The lipid binding properties of HS1BP3 are now better defined. The lipid analysis is very well done but it is done in whole fed cells (as far as is clear from the Methods) the major effects of HS1BP3 are in starved conditions which must produce an altered lipid profile. The inhibitor studies on the lipid enzymes are done in starvation which can not really be correlated with the lipid analysis done in fed cells.
4. The data does not uncover how the loss of HS1BP3 alters PA levels and impacts on the composition or formation of the phagophore.

Reviewer #3 (Remarks to the Author)

I reviewed the original version of this manuscript as reviewer 3, I am completely content with the changes made by the authors and their responses to my concerns and criticisms and I regard the manuscript as acceptable for publication without any change.

Rebuttal NCOMMS-16-13893-T (Simonsen)

Reviewers' comments:

Reviewer #1 (Remarks to the Author):

The authors have responded comprehensively to the reviews. Proper controls and additional for experiments were included to strengthen the authors' data and provide a strong argument.

We thank the reviewer for these encouraging comments.

Reviewer #2 (Remarks to the Author):

The formation of autophagosomes is a complex process which requires a complex machinery and multiple steps, involve both lipids, proteins and interdependencies between lipids and proteins. Holland et al have addressed the role of a lipid binding protein, HS1BP3, which binds PA. HS1BP3 is known to bind HS1 or cortactin. The authors demonstrate that HS1BP3 is a negative regulator of starvation-induced autophagy, which colocalizes with Atg16L1. HS1BP3 is proposed to act by inhibiting PLD1 activity on ATG16L1-positive and TfR-positive membranes. In addition, loss of HS1BP3 increases the synthesis of particular PA species in cells. The authors propose a model whereby HS1BP3 regulates PLD1's access to the Atg16L1, TfR-positive membranes thereby controlling the lipid composition, in particular PA levels of autophagosome precursor membranes. This is a revised version of a manuscript submitted to NCB. The authors have addressed some of the points raised in the previous review but they have done this primarily by removing large sets of data or replacing all the original problematic data. They have made a effort to identify an antibody to the endogenous protein but this has not provided insight into the mechanism. The manuscript, while much easier to understand, is now largely descriptive and provides no definitive mechanism to understand how HS1BP3 works to alter PA levels.

We thank the reviewer for the thorough review of our manuscript. We understand that the reviewer still has some questions as to the exact mechanism by which HS1BP3 works to alter PA levels and autophagy. We clearly show that HS1BP3 regulates PLD1 activity and the localization of PLD1 to ATG16L1 positive autophagosome precursor membranes. We propose that HS1BP3 competes with PLD1 for binding to autophagosome precursor membranes as both proteins contain a PX domain with similar lipid-binding specificities and colocalize with ATG16L1. In this revised version of the manuscript we include new data which strengthen this model as described in more detail below.

Other major points:

1. What are the ATG16L- TfR- positive membranes? How do the authors believe they contribute to autophagosome formation if they only contain ATG16L1 and no up or downstream Atg proteins.

We and others have previously shown that recycling endosomes contain ATG16L1 as well as other ATG proteins and that they provide membrane for autophagosome formation. The Rubinsztein lab has reported that fusion of ATG9- and ATG16L1-containing vesicles with recycling endosomes correlate with autophagosome formation (Puri et al, Cell 2013). The Tooze lab has published that ULK1 and ATG9 are found on Rab11- and transferrin receptor (TfR)-positive recycling endosomes and that amino acid starvation causes TfR and transferrin to localize to forming autophagosomes (Longatti et al, J Cell Biol, 2012). We have shown that the PX-BAR protein SNX18 is a positive regulator of autophagy which facilitates tubulation of

recycling endosomes (TfR and Rab11 positive) membrane containing ATG16L1 (Knævelsrud et al, J Cell Biol 2013). Recently, the Yoshimori lab published that trafficking of ATG9A through the recycling endosomes is an essential step for autophagosome formation (Imai et al, J Cell Sci, 2016).

To further characterize the HS1BP3 and PLD1 positive ATG16L1/TfR compartments shown in this manuscript we have now performed additional experiments (confocal and live cell imaging) which show that:

1) Both overexpressed and endogenous HS1BP3 colocalize with endogenous ATG9 in HEK293 and U2OS cells (new Fig. 3b,c).

2) The HS1BP3-ATG9 positive membranes colocalize with TfR in U2OS (new Fig. 3b).

3) Live cell imaging; HEK293A cells expressing GFP-HS1BP3, GFP-PLD1 or GFP-ATG16L1 together with mCherry-LC3 were starved for 1h in EBSS containing transferrin-Alexa Fluor 647 conjugate and imaged live with Zeiss LSM710 confocal microscope. Still image frames from live scan (new Supplementary Fig. 6) and Movies show that GFP-HS1BP3/-PLD1/-ATG16L1 spots containing Transferrin-Alexa647 traffic to and fuse with mCherry-LC3 positive compartments, further indicating that recycling endosomes provide membrane for autophagosome biogenesis.

Taken together, we conclude that the HS1BP3 and PLD1 positive ATG16L1/TfR structures observed in this manuscript are the same as the ATG16L1-ATG9-ULK1-TfR positive structures observed by others and that these structures are involved in autophagosome biogenesis.

2. The lipid binding properties of HS1BP3 are now better defined. The lipid analysis is very well done but it is done in whole fed cells (as far as is clear from the Methods) the major effects of HS1BP3 are in starved conditions which must produce an altered lipid profile. The inhibitor studies on the lipid enzymes are done in starvation which cannot really be correlated with the lipid analysis done in fed cells.

We apologies that it is not clear from the text (or the methods section) that the lipidomics analysis of control and HS1BP3 depleted cells indeed was done with cells treated with amino acid starvation. This has now been corrected in the text (p9), methods (p22) and figure legend (p 31).

We agree that it is important to use the same conditions when comparing lipid analysis and inhibitor studies, which is the case in this study. We would however like to point out that HS1BP3 depletion increases autophagy both in fed and starved conditions and it is likely that HS1BP3 also in fed cells functions by inhibition of PLD1 activity.

4. The data does not uncover how the loss of HS1BP3 alters PA levels and impacts on the composition or formation of the phagophore.

We agree with the reviewer that we have not shown exactly how HS1BP3 regulates PLD1 activity. In the previous version of this manuscript we showed that loss of HS1BP3 leads to increased PLD1 activity (more PA), increased colocalization of PLD1 with ATG16L1 and increased autophagy. In contrast, we found that overexpression of HS1BP3 reduced the colocalization of PLD1 with ATG16L1. We were not able to detect an interaction between HS1BP3 and PLD1 in co-immunoprecipitation experiments. Thus, we proposed that HS1BP3 and PLD1 compete for binding to ATG16L1 positive structures as both proteins have a PX domain with similar lipid binding specificities.

In order to get some further mechanistic insight into how HS1BP3 regulates PLD1 access to the Atg16L membranes cells we have performed several new experiments;

1) We have repeated and extended the co-immunoprecipitation experiments using endogenous or overexpressed HS1BP3 in combination with endogenous or overexpressed PLD1, but are not able to detect any interaction between the two proteins (data not shown).

2) To further test the hypothesis that the two proteins compete for binding to ATG16L1 positive membranes, we have transfected cells with GFP-HS1BP3 full length, PX or delta-PX constructs and investigated the co-localization of HA-PLD1 with endogenous ATG16L1 in the transfected cells. Interestingly, we find that while cells expressing GFP-HS1BP3 full-length or the PX domain show a significant reduction in the extent of co-localization between HA-PLD1 and ATG16L1 compared to GFP control cells, this is not the case in cells expressing GFP-HS1BP3 delta-PX (new Fig. 6d). As HS1BP3 and PLD1 contain PX domains with similar lipid binding specificity we conclude that the HS1BP3 PX domain prevents recruitment of PLD1 to ATG16L1 positive membranes.

3) To test the PX domain competition hypothesis more directly we tried to set up liposome experiments where the idea was to see if HS1BP3-PX (or full-length), but not delta-PX, could outcompete PLD1's binding to PA or PIPs (or vice versa). We did however encounter problems with this approach; i) Our attempts to purify full length PLD1, from bacteria or mammalian suspension cells, gave protein of unsatisfactory yield and purity. Within the given timeframe we were not successful in setting up other approaches for protein purification, like production in insect cells. Thus, the product we were able to produce was not useful for our assays as this would require much more protein. ii) As already mentioned, the membrane interaction of HS1BP3 and PLD1 is not only governed by PA but is also affected by the presence of different PIPs. A proper experimental approach would therefore include liposomes with different PIP-compositions, increasing the sample number and again raising the need for high protein yield. Because of these issues we were unable to further pursue our in vitro studies any further in this direction.

Reviewer #3 (Remarks to the Author):

I reviewed the original version of this manuscript as reviewer 3, I am completely content with the changes made by the authors and their responses to my concerns and criticisms and I regard the manuscript as acceptable for publication without any change.

We thank the reviewer for this positive feedback.

Reviewers' Comments:

Reviewer #2 (Remarks to the Author)

The authors have clearly done their best to address the concerns and in light of the time commitment required for unresolved issues I accept their comments and recommend publication.